# Efficient Homogenization-Ultrasound-Assisted Extraction of Anthocyanins and Flavonols from Bog Bilberry (*Vaccinium uliginosum* L.) Marc with Carnosic Acid as an Antioxidant Additive

**DOI:** 10.3390/molecules24142537

**Published:** 2019-07-11

**Authors:** Yusong Jin, Yunhui Zhang, Dongmei Liu, Dewen Liu, Chunying Zhang, Huijuan Qi, Huiyan Gu, Lei Yang, Zhiqiang Zhou

**Affiliations:** 1Key Laboratory of Forest Plant Ecology, Ministry of Education, Northeast Forestry University, Harbin 150040, China; 2Bureau of Silviculture in Daxing’anling Region, Jiagedaqi, Heilongjiang 165000, China; 3Heilongjiang Institute of Construction Technology, Harbin 150025, China; 4Heilongjiang Daxing’anling Academy of Agriculture and Forestry Sciences, Jiagedaqi 165000, China; 5School of Forestry, Northeast Forestry University, Harbin 150040, China

**Keywords:** antioxidant capacity, polyphenol, fruit marc, natural antioxidant, response surface methodology

## Abstract

To explore the optimum conditions for the extraction of anthocyanins and flavonols from bog bilberry (*Vaccinium uliginosum* L.) marc on a single-factor experimental basis, a response surface methodology was adopted for this intensive study. The extraction procedure was carried out in a Waring blender and followed an ultrasonic bath, and the natural antioxidant carnosic acid was added to inhibit oxidation. The optimum extraction conditions were as follows: a volume fraction of ethanol of 70%, an antioxidant content of 0.02% (the mass of sample) carnosic acid, a liquid–solid ratio of 16 mL/g, a homogenization time of 3 min, a reaction temperature of 55 °C, an ultrasound irradiation frequency of 80 kHz, an ultrasound irradiation power of 200 W, and an ultrasound irradiation time of 40 min. Satisfactory yields of anthocyanins (13.95 ± 0.37 mg/g) and flavonols (3.51 ± 0.16 mg/g) were obtained. The experimental results showed that the carnosic acid played an effective antioxidant role in the extraction process of anthocyanins and flavonols with a green and safety guarantee.

## 1. Introduction

*Vaccinium uliginosum* L. (bog bilberry) is a deciduous shrub that is classified as belonging to the Ericaceae family and genus *Vaccinium*. Modern food chemistry and pharmacological studies have shown that the bog bilberry contains a number of phytochemicals that are beneficial to the human body, such as anthocyanins and flavonols [1]. They are both important phenolic compounds that have many physiologically active functions such as the free scavenging [2], antibacterial and antifungal activity [3], and the protection of vision [4]. Because of various healthy uses, bog bilberries have long received public recognition and are cultivated on a massive commercial scale all over the world.

In general, the fresh bog bilberry is always used to make beverages and fruit wine. Mechanical crushing is often used to extract juice from the berries [5], and the fruit marc is usually discarded as waste. Unfortunately, a great number of compounds that are good for human health, such as anthocyanins and flavonols, tend to remain in high concentrations in the marc [6,7]. Thus, the extraction and separation of anthocyanins and flavonoids from fruit marc should be paid enough attention. The anthocyanins in bog bilberries are more complex than those present in some common edible berries (such as raspberry [8], haskap berry [9], and blackcurrant [10]); as regards to their structure, the aglycone part is composed of malvidin, delphinidin, petunidin, cyanidin, and peonidin, while the glycosyl part is composed of glucoside, galactoside, arabinoside, and xyloside [1]. At the same time, bog bilberry is also rich in flavonols such as hyperin, isoquercitrin, myricetin, and quercetin [1]. However, the existence of the glycosidic bonds in anthocyanins and flavonols is easily destroyed by enzymes such as glucosidase and cellulase in plants, and phenolic hydroxyl groups tend to be oxized in the molecules of anthocyanins and flavonols [11]. Moreover, the anthocyanins and flavonols in the bog bilberry marc may also be isomerized after long-term heating and concentration [12]. Because of these characteristics, some developed techniques, including the inhibition of enzymatic oxidation by microwave treatment [13] and the addition of synthetic antioxidants to the extraction solvent [10], were adopted for extraction of anthocyanins and flavonols. However, microwave extraction is difficult to scale up to realize commercial production because of the absence of large-scale equipment, hence the treatment capacity is limited. In several countries, governments have recognized that synthetic antioxidants have certain adverse effects on human health; thus, the use of synthetic antioxidants has been restricted [14].

The Labiatae family is usually regarded as a significant source of natural antioxidant [15]. Rosmarinus officinalis is a shrub of Labiatae family. Its two main constituents, carnosic acid and rosmarinic acid, are also be considered as natural antioxidants. Carnosic acid (CA) is a natural and relatively healthy phenolic diterpene antioxidant isolated from rosemary [16]. As CA is a fat-soluble natural antioxidant, its antioxidant effect is better than that of synthetic antioxidants in oil-containing products. Rosmarinic acid (RA), a water-soluble natural antioxidant, has also been used to avoid oxidation of target components [17]. CA and RA have been recognized as food additives [18] and are increasingly being used in the food, nutrition, health [19], and cosmetics industries [18].

Independent of the extraction method, mechanical crushing, heating, and drying are usually required before the extraction operation of fruit marc. However, fruit marc is usually massive solid, which is liable to produce dust pollution during crushing compared with fruit berry. In contrast to conventional operations, homogenization processing greatly increases efficiency by smashing the raw material in a suitable solvent through high-speed mechanical shearing without excessive heat and pressure. The raw material can be sheared, fluidized thoroughly, and then the active compounds dissolved in the solvent to its utmost. Homogenization processing has exerted a positive influence on obtaining natural colorings [20], flavonoids [21], polysaccharides [22], and fruit oils [23]. Ultrasound-assisted extraction (UAE) has the advantages of high efficiency and reproducibility, short extraction time, low solvent consumption, simple manipulation, and high level of automation [24,25,26,27]. The ultrasound mechanism is conducive to greatly improving the solvent infiltration into cellular materials, and the cavitation effects promote the release of cell inclusion into the bulk media [28,29]. UAE has now become an environmentally friendly method to expand commercial production.

Among a number of influencing factors considered in the extraction of fruit marc, reducing the impact on the environment and energy consumption has gradually become the focus. To meet the overall demands of industrial-scale marc extraction, new extraction methods need to be developed. Several studies concerning the extraction of anthocyanins and flavonols from bog bilberry exist in literature [30,31], however, to the best of our knowledge, the optimized extraction process of polyphenols from this matrix with the addition of a natural antioxidant has never been reported. Therefore, an improved homogenization-ultrasound-assisted extraction (HUAE) method with the addition of carnosic acid, as natural antioxidant, was adopted in this paper. The main influence parameters, including the ethanol concentration; types and dose of antioxidants; liquid–solid ratio; homogenization time; temperature; and ultrasound irradiation frequency, power, and time, were optimized systematically by the univariate method, and the temperature and ultrasound irradiation power and time were selected for further optimization through the response surface methodology.

## 2. Results and Discussion

### 2.1. HPLC Chromatograms for Anthocyanins and Flavonols

The chromatogram of anthocyanins obtained from 70% ethanol extract is shown in Figure 1a, while that of the four flavonols obtained is shown in Figure 1b. The elution order was hyperin, isoquercitrin, myricetin, and quercetin, respectively. The linear equations for the calibration curve of hyperin, isoquercitrin, myricetin, and quercetin were as follows: *Y_Hyperin_* = 42.372*x* + 56.367 (*R*^2^ = 0.9994, *n* = 7); *Y_Isoquercitrin_* = 57.534*x* + 66.025 (*R*^2^ = 0.9999, *n* = 7); *Y_Myricetin_* = 45.622*x* + 118.45 (*R*^2^ = 0.9998, *n* = 7); and *Y_Quercetin_* = 100.59*x* – 136.64 (*R*^2^ = 0.9999, *n* = 7); these equations were graphed by plotting the peak area (*Y*) against the concentration (*x*). The calibration curves showed good linearity for hyperin, isoquercitrin, myricetin, and quercetin, ranging from 0.01 to 1.00 mg/mL.

### 2.2. Influence of the Ethanol Concentration

As shown in Figure 2A,B, the yields of anthocyanins and flavonols extracted with ethanol at different concentrations were higher than that extracted only with ultrapure water under the same conditions (solvent volume: 30 mL, liquid–solid ratio: 10 mL/g, CA dosage: 0.02%, homogenization time: 3 min, temperature: 50 °C, ultrasound frequency: 45 kHz, ultrasound irradiation power: 200 W, and ultrasound irradiation time: 30 min). When the concentration of ethanol increased from 0% to 90%, the yields of anthocyanins and flavonols increased first and then decreased with the increasing concentration of ethanol. At 70% of ethanol concentration, the yields of anthocyanins and flavonols reached the maximum of 8.01 ± 0.30 mg/g and 1746.55 ± 48.90 μg/g, respectively. Therefore, 70% of ethanol concentration was selected.

### 2.3. Influence of the Antioxidant Species

Figure 2C,D showed the changes in the yields of anthocyanins and flavonols relative to the blank control as affected by the addition of the six different antioxidants (other extraction conditions: fixing 30 mL of solvent volume, 50% of ethanol concentration, 0.02% of antioxidant dosage, 10 mL/g of liquid–solid ratio, 3 min of homogenization time, 50 °C of temperature, 45 kHz of ultrasound irradiation frequency, 200 W of ultrasound irradiation power, and 30 min of ultrasound irradiation time). Regardless of which antioxidant species was added, both yields were higher than that of the blank control with significant difference. Moreover, there were also significant differences among the other four antioxidants on the yield of anthocyanins except for *tert*-butylhydroquinone (TBHQ) and CA + RA. The anthocyanins and flavonols reached the highest yields of 6.52 ± 0.27 mg/g and 1226.45 ± 38.02 μg/g, respectively, at 0.02% of CA. Thus, CA was used for subsequent optimization.

### 2.4. Influence of the Dosage of CA

In Figure 2E,F, the yields of anthocyanins and flavonols show a significant increment when the CA dosage was increased from 0% to 0.02% (other extraction conditions: 30 mL of solvent volume, 50% of ethanol concentration, 10 mL/g of liquid–solid ratio, 3 min of homogenization time, 50 °C of temperature, 45 kHz of ultrasound irradiation frequency, 200 W of ultrasound irradiation power, and 30 min of ultrasound irradiation time). Although the yields of anthocyanins and flavonols enhanced with the increasing of CA dosage from 0.02% to 0.04%, the difference was not significant. Therefore, 0.02% of CA dosage was chosen.

### 2.5. Single-Factor Extraction Experiments

#### 2.5.1. Influence of Liquid–Solid Ratio

When the liquid to solid ratio was greater than 16 mL/g, the effect on the yields of anthocyanins and flavonols was not significant, as shown in Figure 3A,B (other extraction conditions: 30 mL of solvent volume, 70% of ethanol concentration, 0.02% of CA dosage, 3 min of homogenization time, 50 °C of temperature, 45 kHz of ultrasound irradiation frequency, 200 W of ultrasound irradiation power, and 30 min of ultrasound irradiation time). That is, 16 mL/g of liquid–solid ratio almost completely dissolved the effective components, and the yields of anthocyanins and flavonols were 10.55 ± 0.37 mg/g and 2137.39 ± 61.98 μg/g, respectively. Considering the energy consumption and production cost, a liquid–solid ratio of 16 mL/g was chosen for later experiments.

#### 2.5.2. Selection of Homogenization Time

The homogenization time determined the degree of cell fragmentation, which in turn affected the mass transfer rate of the anthocyanins and flavonols in the extraction process [32]. In Figure 3C,D, the yields of anthocyanins and flavonols increased first and then decreased with a prolonged homogenization time (other extraction conditions: 30 mL of solvent volume, 50% of ethanol concentration, 0.02% of CA dosage, 10 mL/g of liquid–solid ratio, 50 °C of temperature, 45 kHz of ultrasound irradiation frequency, 200 W of ultrasound irradiation power, and 30 min of ultrasound irradiation time). It could be speculated that the thermal effect of the high-speed homogenization might result in degradation or isomerization in anthocyanins and flavonols with relatively poor stability [10]. Above all, the optimum homogenization time was 3 min.

#### 2.5.3. Influence of Temperature

In Figure 3E,F, the higher the temperature was, the greater the yields of anthocyanins and flavonols with a significant difference (other extraction conditions: 30 mL of solvent volume, 70% of ethanol concentration, 0.02% of CA dosage, 16 mL/g of liquid–solid ratio, 3 min of homogenization time, 80 kHz of ultrasound irradiation frequency, 200 W of ultrasound irradiation power, and 30 min of ultrasound irradiation time). Perhaps the extraction efficiency was improved by the high temperature in the UAE, increasing of the number of cavitation bubbles [33]. However, the high temperature could also cause oxidation, degradation, or isomerization of the target compounds, resulting in reduced yields. Because the upper rated temperature of the ultrasonic machine was 60 °C, a temperature range of 40–60 °C was chosen for subsequent optimization.

#### 2.5.4. Influence of Ultrasound Irradiation Frequency

Figure 4A,B showed that the order of the yields of anthocyanins and flavonols from high to low was 80 kHz > 45 kHz > 100 kHz using the three fixed frequencies of the ultrasonic bath (other extraction conditions: 30 mL of solvent volume, 70% of ethanol concentration, 0.02% of CA dosage, 16 mL/g of liquid–solid ratio, 3 min of homogenization time, 50 °C of temperature, 200 W of ultrasound irradiation power, and 30 min of ultrasound irradiation time). Maybe the intensity of cavitation effects of the ultrasound was directly related to its frequency; the higher the frequency was, the smaller the cavitation bubbles and the weaker the cavitation intensity [34]. Because the inherent thermal effect of the ultrasound would improve with the increasing frequency [35], to prevent the adverse effect of the temperature rise on the yields, it was advisable to select 80 kHz as the optimized ultrasound irradiation frequency.

#### 2.5.5. Influence of Ultrasound Irradiation Power

The ultrasound irradiation power had a direct effect on cell breakage and the entrance of solvent into cells, thereby affecting the yields of anthocyanins and flavonols. In Figure 4C,D, the yields continued to grow with the increasing ultrasound irradiation power (other extraction conditions: 30 mL of solvent volume, 70% of ethanol concentration, 0.02% of CA dosage, 16 mL/g of liquid–solid ratio, 3 min of homogenization time, 50 °C of temperature, 45 kHz of ultrasound irradiation frequency, and 30 min of ultrasound irradiation time). This pattern might be explained by the process of numerous bubbles forming in the solvent and rapidly imploding, accelerating the cell wall rupture and solvent infiltration with the increase of ultrasound irradiation power. The yields reached a high level when the ultrasound irradiation power was 200 W within the parameter limits of the ultrasonic machine. Therefore, the optimized ultrasonic irradiation power range was determined to be 120–200 W.

#### 2.5.6. Influence of Ultrasound Irradiation Time

The ultrasound irradiation time was also an important factor in the ultrasound extraction. With the extension of ultrasound irradiation time, the cumulative effects of ultrasonic cavitation and thermal and mechanical effects were generally enhanced [36]. Figure 4E,F showed that the yields of anthocyanins and flavonols increased rapidly when the ultrasound irradiation time was prolonged from 0 to 50 min, and then the yields decreased with the continued extending time (other extraction conditions: fixing 30 mL of solvent volume, 70% of ethanol concentration, 0.02% of CA dosage, 16 mL/g of liquid–solid ratio, 3 min of homogenization time, 60 °C of temperature, 80 kHz of ultrasound irradiation frequency, and 200 W of ultrasound irradiation power). It was very possible that there was a large concentration difference between the inside and outside of the cells, which led to anthocyanins and flavonols dissolving rapidly from cells in the early stage of the ultrasound extraction process. However, the concentration difference decreased with the prolongation of the ultrasound irradiation time, generated by the ultrasound damaging the structure of anthocyanins and flavonols, leading to decreases in the yields. This result was similar to Tiwari et al. [37]. Therefore, the best ultrasound irradiation time range was determined to be 30–50 min.

### 2.6. Parameter Optimization by Response Surface Methodology

#### 2.6.1. Experimental Design and Statistical Analysis

To obtain the maximum amount of anthocyanins and flavonols from bog bilberry marc, on the basis of the single factor experiments and the Plackett–Burman design, the ultrasound irradiation power, temperature, and ultrasound irradiation time were selected as the major factors. The yields of anthocyanins and flavonols were chosen as the response values by fixing 30 mL of solvent volume, 70% of ethanol concentration, 0.02% of CA dosage, 16 mL/g of liquid–solid ratio, 3 min of homogenization time, and 80 kHz of ultrasound irradiation frequency. A quadratic regression equation of three factors and three levels was designed to fit the functional relationship between the factors and indexes (response values). The levels of the experimental factors, design, and results are shown in Table 1.

Each of these experiments was performed three times, and the average of the three experimental results was taken as the corresponding response value. The experimental data were analyzed by Design-Expert 8.0.6 software using quadratic regression analysis, and the following equations were obtained:*Y_Anthocyanins_* = 12.50 + 2.06*X_1_* + 0.69*X_2_* – 0.074*X_3_* – 0.24*X_1_X_2_* – 0.28*X_1_X_3_* – 0.022*X_2_X_3_* – 0.66*X_1_*^2^ – 0.63*X_2_*^2^ – 0.55*X_3_*^2^,
*Y_Flavonols_* = 3.29 + 0.56*X_1_* + 0.24*X_2_* + 0.29*X_3_* – 0.032*X_1_X_2_* – 0.18*X_1_X_3_* – 0.025*X_2_X_3_* – 0.40*X_1_*^2^ – 0.19*X_2_*^2^ – 7.50 × 10^-4^*X_3_*^2^,
where *Y_Anthocyanins_* is the total yields of anthocyanins; *Y_Flavonols_* is the total yield of hyperin, isoquercitrin, myricetin, and quercetin; and *X_1_*, *X_2_*, and *X_3_* are the ultrasound irradiation power (W), temperature (°C), and ultrasound irradiation time (min), respectively. From the analysis of variance, it could be seen that the models reached extremely significant levels (*p* < 0.0001). “Lack of fit” represented the probability that the predicted values did not fit the experimental values of a model. In Table 2, the lack of fit values of anthocyanins and flavonols are 0.3187 and 0.6768, respectively (> 0.05), indicating that the lack of fit of the models of anthocyanins and flavonols was not significant. Therefore, such models were suitable for use without the introduction of high degree functional terms. Meanwhile, the correlation coefficient (*R*^2^) of the anthocyanins and flavonols equations were 0.9925 and 0.9831 (>0.98), respectively, showing that the models had a high degree of applicability.

Moreover, the coefficient of variation (CV%) indicated the reproducibility and credibility of a model [38]. The smaller the CV% value, the higher the credibility of a model, and vice versa. The coefficients of variation in this experiment were 1.87% and 3.54%, respectively, indicating that the experimental operation was credible. The models were in accordance with the experimental results of Agil et al. [39] and Fan et al. [40].

#### 2.6.2. Analysis of the Interaction Effects

With regard to the interaction between the relevant variables and the optimization of the extraction conditions, the factors of the ultrasound irradiation power (*X_1_*), temperature (*X_2_*), and ultrasound irradiation time (*X_3_*) were reflected in the response surfaces (Figure 5). The interaction between the ultrasound irradiation power (*X_1_*) and temperature (*X_2_*) relative to the yields of anthocyanins and flavonols is shown in Figure 5a,d with a fixed ultrasound irradiation time (0 level). As the ultrasound irradiation power (*X_1_*) increased, the yield of anthocyanins continued to increase dramatically; however, the yield of flavonols first increased, followed by a small dip. Compared with the ultrasound irradiation power (*X_1_*), the effect of increasing the temperature (*X_2_*) on the yields of anthocyanins and flavonols was not as great, showing a trend of rising first and maintaining stability after that. Figure 5b,e present the effects of the ultrasound irradiation power (*X_1_*) and time (*X_3_*) on the yields of anthocyanins and flavonols at a constant temperature (0 level). With the increasing of ultrasound irradiation power (*X_1_*), the yields of both anthocyanins and flavonols continued to increase. Figure 5e shows an ever-increasing trend of the flavonols yield as the ultrasound irradiation time (*X_3_*) was prolonged, while the anthocyanin yield increased first and fell afterwards. Figure 5c,f show the interaction of the temperature (*X_2_*) and the ultrasound irradiation time (*X_3_*) in the case of a fixed ultrasound irradiation power (*X_1_*). The yield of anthocyanins increased first and then remained steady when the temperature(*X_2_*) was increased. However, it increased and then decreased with the increasing ultrasound irradiation time (*X_3_*). This trend is similar to that of Fang et al. in ultrasonic extraction of pristimerin from *Celastrus orbiculatus* [41]. As far as the yield of flavonols is concerned, the change was not significant with the interaction of reaction temperature (*X*_2_) and ultrasound irradiation time (*X_3_*). It can be presumed that the yield of flavonols was insensitive to the combined effects of temperature and ultrasound irradiation time.

Thus, the optimum extraction conditions were as follows: *X_1_*, 197.22 W; *X_2_*, 54.60 °C; and *X_3_*, 40.43 min. (then approximated to 200 W, 55 °C, and 40 min, respectively, in order to simplify the validation tests applicability). The total theoretical yields of anthocyanins and flavonols under the best extracting conditions were 13.91 mg/g and 3.53 mg/g, respectively.

#### 2.6.3. Validation Tests

We performed three verification tests to verify the results and the reliability of response surface models. The actual yields of anthocyanins and flavonols were 13.95 ± 0.37 mg/g and 3.51 ± 0.16 mg/g, respectively, which satisfied the theoretical yields of anthocyanins and flavonols.

## 3. Materials and Methods

### 3.1. Materials and Chemicals

Mature bog bilberry was harvested from the Jiagedaqi Forestry Bureau in Great Xing’an Mountains of Heilongjiang Province (northeast of China) in May 2018, and authenticated by Prof. Baojiang Zheng from the College of Life Sciences, Northeast Forestry University, China. The voucher specimen was deposited in the herbarium of Northeast Forestry University. The fruit marc was obtained after squeezing the berries, and the marc was stored at −4 °C before starting the extraction. The reference substances cyanidin-3-*O*-d-glucoside, hyperin, isoquercitrin, myricetin, and quercetin were purchased from Sigma-Aldrich (Shanghai, China). Rosmarinic acid, carnosic acid, potassium chloride, and sodium acetate were purchased from Aladdin (Shanghai, China). The synthetic antioxidants *tert*-butylhydroquinone (TBHQ), butylated hydroxytoluene (BHT), and butylated hydroxyanisole (BHA) were obtained from Sinopharm Chemical Reagent Co., Ltd. (Beijing, China). Acetonitrile, methanol, formic acid, and phosphoric acid of chromatographic grade were purchased from Thermo Fisher Scientific (Shanghai, China). Ultrapure water was acquired from a Milli-Q purification apparatus (Bedford, MA, USA). All fluid samples were filtered through a 0.45 μm nylon filter membrane before HPLC analysis.

### 3.2. Experimental Apparatus

The homogenization processing was conducted in a 570 mL JZM-3001 food-grade blender (Guangdong, China). The ultrasound extraction was performed in a KQ-200 VDB triple computerized numerical control (CNC) ultrasonic bath (Jiangsu, China), and the working frequencies were three fixed grades of 45, 80, and 100 kHz. The ultrasonic bath was a rectangular container (300 mm × 150 mm × 150 mm) that had 50 kHz transducers annealed at the bottom. The maximum output of the ultrasound irradiation power was 200 W, and the power was continuously adjustable on the scale of 80–200 W. The inlet and outlet water was displaced to control the temperature.

### 3.3. HUAE Procedure

The bog bilberry marc and the antioxidant were weighed, and their mixture was dissolved in aqueous ethanol, then the solid–liquid mixture was homogenized in the blender for a period of time. A 150 mL Erlenmeyer flask was used to contain the suspension that was subjected to the ultrasound treatment (under various ultrasonic conditions). Under a series of extraction conditions, we conducted each experiment three times.

A Box–Behnken design was used for optimizing the HUAE process with respect to three variables. Each experiment was repeated three times, and 17 experimental data sets were analyzed using analysis of variance (ANOVA). The fitting equation of the response surface regression was as follows:(1)$$Y=\beta_{0}+\sum_{i=1}^{3}\beta_{i}X_{i}+\sum_{i=1}^{3}\beta_{ii}X_{i}^{2}+\sum_{i=1}^{2}\sum_{j=i+1}^{3}\beta_{ij}X_{i}X_{j},$$

In Equation (1), *Y* represents the estimated response of total anthocyanins and flavonols; *β_0_*, *β_i_*, *β_ii_*, and *β_ij_* are the regression coefficients, which denote the intercept, linear, quadratic, and interaction terms, respectively; and *X_i_* and *X_j_* are coded independent variables. Design-Expert (Version 8.0.6) software was applied for the regression analysis.

### 3.4. HPLC Analysis

The HPLC analysis was performed with an Agilent 1260 HPLC system equipped with an autosampler, a G1311C model pump, and a G1314B UV detector. The separation process was performed using an Agilent-Extend-C18 column (4.6 × 250 mm, 5 μm, Palo Alto, CA, USA). For the qualitative analysis of anthocyanins, 1% formic acid in pure water (A) and 100% acetonitrile (B) were selected as the elution solvents and used at a flow rate of 1 mL/min. The elution gradient was as follows: a linear gradient from 5% to 18% B, 0–30 min; a linear gradient from 18% to 25% B, 30–40 min; and a linear gradient from 25% to 5% B, 40–60 min. The injection volume was 10 μL at a column temperature of 25 °C with a detection wavelength of 525 nm (the apparatus and the HPLC column were flushed with acetonitrile or methanol, and then the column was balanced with mobile phase until the baseline of the chromatogram getting stable before each injection).

For the qualitative and quantitative analysis of the flavonols, the mobile phase consisted of deionized water with phosphoric acid (100:1, v/v) (A) and 100% methanol (B). The flow rate was 0.9 mL/min. The injection volume of each sample was 10 μL. The gradient elution program was as follows: a linear gradient from 25% to 30% B, 0–25 min; a linear gradient from 30% to 37.5% B, 25–35 min; a linear gradient from 37.5% to 70% B, 35–45 min; and a linear gradient from 70% to 25% B, 45–60 min. The column temperature was retained at 30°C, and an ultraviolet detector was set at 360 nm. The chromatographic peaks of the analytes were confirmed and identified by comparing their relative retention times and UV spectra with those of the standard compounds. Equation (2) was used to quantify the yields of flavonols:(2)$$\mathrm{Yield}\;(\mathrm{mg}/g)=\frac{\mathrm{mean}\;\mathrm{mass}\;\mathrm{of}\;\mathrm{hyperin},\;\mathrm{isoquercitrin},\;\mathrm{myricetin}\;\mathrm{and}\;\mathrm{quercetin}\;\mathrm{in}\;\mathrm{samples}\;(\mathrm{mg})}{\mathrm{mean}\;\mathrm{mass}\;\mathrm{of}\;\mathrm{the}\;\mathrm{samples}\;(g)}$$

The mean mass of hyperin, isoquercitrin, myricetin, and quercetin was confirmed with three samples using HPLC analysis, and the average mass of samples was considered as the mean mass of bog bilberry marc. Each repetition was expressed as mean ± SD (standard deviation), and the average value of each treatment was compared by variance analysis (using SPSS 17.0). Duncan’s test was used to determine the significant difference between the average of parameters (*p* < 0.05).

### 3.5. Quantitative Analysis of Anthocyanins

The classical pH differential method was used to determine the content of anthocyanins in bog bilberry marc extracts [42]. The principle of the pH differential method is diluting the sample liquid to appropriate multiples with the help of pH 1.0 (0.2 mol/L potassium chloride: 0.2 mol/L HCl (25:67, v/v)) and pH 4.5 (1 mol/L sodium acetate: 1 mol/L HCl/H_2_O (10:6:9, v/v/v)) buffers, then placing the sample liquid in the dark and balancing for 15 min. The absorbance of samples was determined at 510 nm and 700 nm in a cuvette, and ultrapure water was used as a blank control. The absorbance of the samples after dilution (ΔA) was calculated according to the following formula:ΔA = (A_510_ − A_700_)_pH 1.0_ − (A_510_ − A_700_)_pH 4.5_(3)

The anthocyanin content in the samples was calculated using the following formula:Anthocyanin content (mg/L) = [(ΔA × Mw)/(ε × d)] × Df × 1000 (4)
where Mw is the relative molecular mass of cyanidin-3-*O*-d-glucoside (484.82 mg/mol); ε is the molar absorption coefficient of cyanidin-3-*O*-d-glucoside (24825 mol^−1^); Df is the dilution factor (the total dilution multiple of the sample); and d is the optical path of the cuvette.

## 4. Conclusions

In this study, an improved homogenization-ultrasound-assisted extraction method with the addition of natural antioxidants was used to extract anthocyanins and flavonols from bog bilberry marc. The method reduced the loss of effective ingredients from raw materials with improved extraction efficiency. The extraction conditions were optimized systematically using the univariate and response surface methods. The addition of CA as a natural antioxidant notably increased the yields of anthocyanins and flavonols, and CA was safer than synthetic antioxidants when used in foods. Response surface analysis of anthocyanins and flavonols presented independent and interactive effects of the variables on the yields of target compounds. The extraction procedures can be applied to extract active components that are easily oxidized from other plant materials according to the experimentally tested factors.

## Figures and Tables

**Figure 1 molecules-24-02537-f001:**
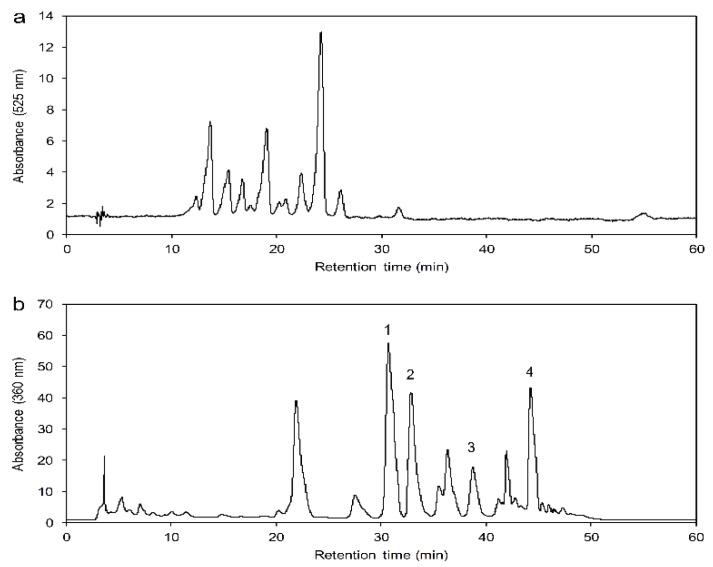
HPLC chromatogram for anthocyanins (**a**) and flavonols (**b**) in a sample extract.

**Figure 2 molecules-24-02537-f002:**
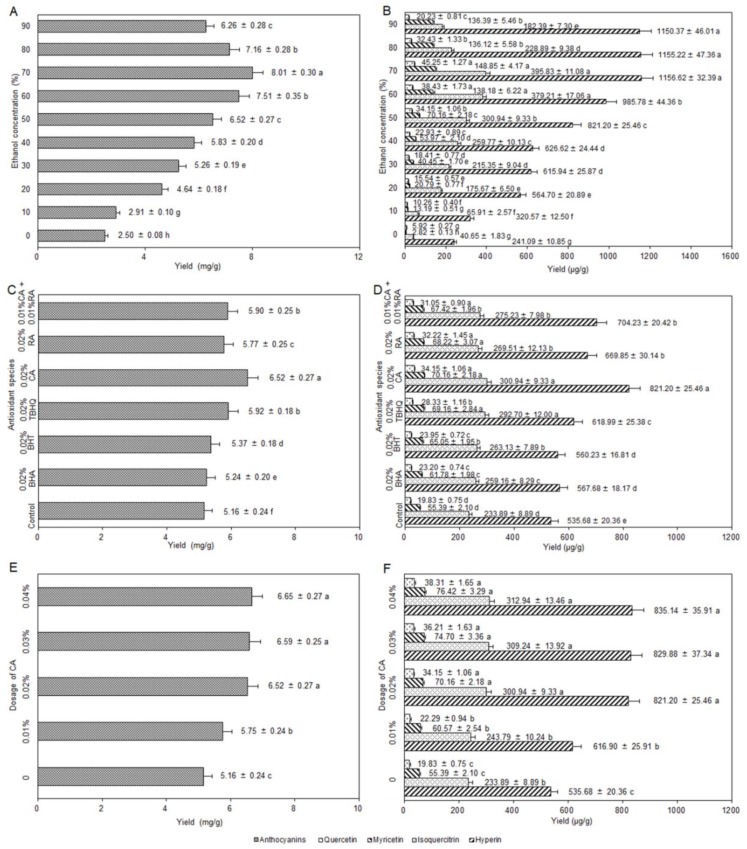
Influence of ethanol concentration on the extraction yields of anthocyanins (**A**) and flavonols (**B**); influence of antioxidant species on the extraction yields of anthocyanins (**C**) and flavonols (**D**); influence of dosage of CA on the extraction yields of anthocyanins (**E**) and flavonols (**F**). CA, carnosic acid; RA, rosmarinic acid; TBHQ, *tert*-butylhydroquinone; BHT, butylated hydroxytoluene; BHA, butylated hydroxyanisole.

**Figure 3 molecules-24-02537-f003:**
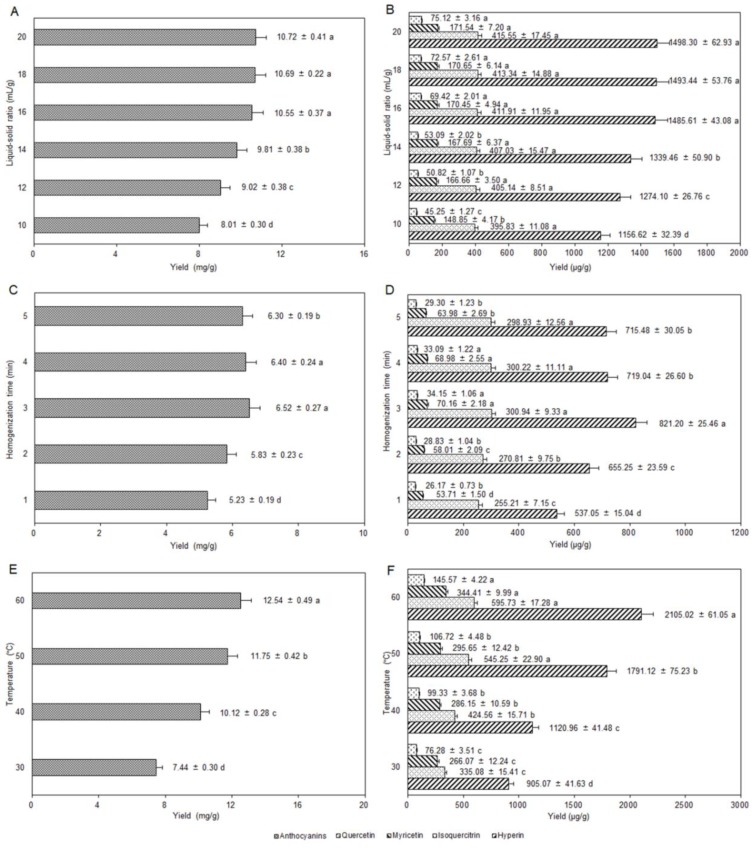
Influence of liquid–solid ratio on the extraction yields of anthocyanins (**A**) and flavonols (**B**); influence of homogenization time on the extraction yields of anthocyanins (**C**) and flavonols (**D**); influence of temperature on the extraction yields of anthocyanins (**E**) and flavonols (**F**).

**Figure 4 molecules-24-02537-f004:**
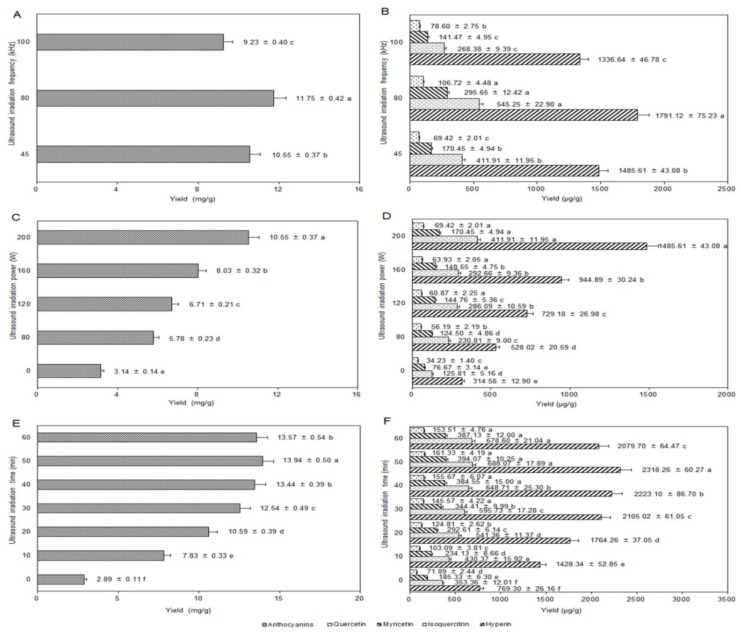
Influence of ultrasound irradiation frequency on the extraction yields of anthocyanins (**A**) and flavonols (**B**); influence of ultrasound irradiation power on the extraction yields of anthocyanins (**C**) and flavonols (**D**); influence of ultrasound irradiation time on extraction yields of anthocyanins (**E**) and flavonols (**F**).

**Figure 5 molecules-24-02537-f005:**
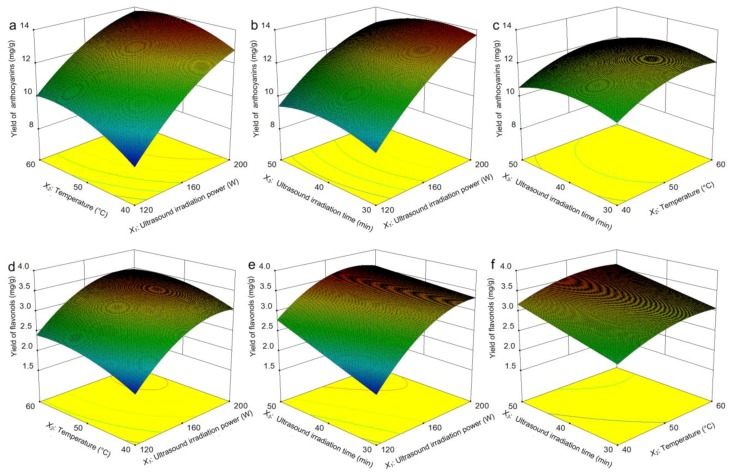
Optimization of the yields of anthocyanins and flavonols by response surface methodology. The effect of the ultrasound irradiation power and temperature on the extraction yields of anthocyanins (**a**) and flavonols (**d**), the effect of the ultrasound irradiation power and ultrasound irradiation time on the extraction yields of anthocyanins (**b**) and flavonols (**e**), the effect of the temperature and ultrasound irradiation time on the extraction yields of anthocyanins (**c**) and flavonols (**f**).

**Table 1 molecules-24-02537-t001:** Box–Behnken design (BBD) for the actual and predicted values of the yields of anthocyanins and flavonols.

Run	Factors			Response 1		Response 2	
	Ultrasound Irradiation Power (W)	Reaction Temperature (°C)	Ultrasound Irradiation Time (min)	Actual Anthocyanins Yield (mg/g)	Predicted Anthocyanins Yield (mg/g)	Actual Flavonols Yield (mg/g)	Predicted Flavonols Yield (mg/g)
1	120 (−1)	40 (−1)	40 (0)	8.18	8.22	1.93	1.87
2	200 (+1)	40 (−1)	40 (0)	12.97	12.81	3.01	3.06
3	120 (−1)	60 (+1)	40 (0)	9.92	10.08	2.46	2.41
4	200 (+1)	60 (+1)	40 (0)	13.77	13.73	3.41	3.47
5	120 (−1)	50 (0)	30 (−1)	9.22	9.02	1.81	1.86
6	200 (+1)	50 (0)	30 (−1)	13.69	13.69	3.39	3.34
7	120 (−1)	50 (0)	50 (+1)	9.43	9.43	2.74	2.79
8	200 (+1)	50 (0)	50 (+1)	12.80	13.00	3.62	3.57
9	160 (0)	40 (−1)	30 (−1)	10.52	10.67	2.55	2.55
10	160 (0)	60 (+1)	30 (−1)	12.07	12.11	3.08	3.08
11	160 (0)	40 (−1)	50 (+1)	10.61	10.57	3.18	3.18
12	160 (0)	60 (+1)	50 (+1)	12.07	11.92	3.61	3.61
13	160 (0)	50 (0)	40 (0)	12.55	12.50	3.34	3.29
14	160 (0)	50 (0)	40 (0)	12.25	12.50	3.31	3.29
15	160 (0)	50 (0)	40 (0)	12.73	12.50	3.20	3.29
16	160 (0)	50 (0)	40 (0)	12.61	12.50	3.16	3.29
17	160 (0)	50 (0)	40 (0)	12.36	12.50	3.46	3.29

**Table 2 molecules-24-02537-t002:** The estimated regression coefficients and analysis of variance (ANOVA) for the quadratic models of anthocyanins and flavonols determined from BBD ^a^.

Source	Sum of Squares		Df	Mean Square		*F*-value		*p*-value	
	AC	FV		AC	FV	AC	FV	AC	FV
Model	43.72	4.64	9	4.86	0.52	102.64	45.17	<0.0001 *	<0.0001 *
*X_1_*	33.95	2.52	1	33.95	2.52	717.25	220.65	<0.0001 *	<0.0001 *
*X_2_*	3.85	0.45	1	3.85	0.45	81.35	39.10	<0.0001 *	0.0004 *
*X_3_*	0.04	0.67	1	0.04	0.67	0.92	58.91	0.3696	0.0001 *
*X_1_X_2_*	0.22	0.00	1	0.22	0.00	4.67	0.37	0.0676	0.5622
*X_1_X_3_*	0.30	0.12	1	0.30	0.12	6.39	10.73	0.0393 *	0.0136 *
*X_2_X_3_*	0.00	0.00	1	0.00	0.00	0.04	0.22	0.8420	0.6541
*X_1_^2^*	1.84	0.68	1	1.84	0.68	38.90	59.95	0.0004 *	0.0001 *
*X_2_^2^*	1.66	0.15	1	1.66	0.15	35.17	13.07	0.0006 *	0.0086 *
*X_3_^2^*	1.29	0.00	1	1.29	0.00	27.28	0.00	0.0012 *	0.9889
Residual	0.33	0.08	7	0.05	0.01				
Lack of fit	0.18	0.02	3	0.06	0.01	1.62	0.55	0.3187	0.6768
Pure error	0.15	0.06	4	0.04	0.01				
Cor total	44.06	4.72	16						
**Credibility Analysis of the Regression Equations**	**Index Mark**	**SD**	**Mean**	**CV%**	**Press**	***R^2^***	**Adjust *R^2^***	**Predicted *R^2^***	**AP**
	AC	0.22	11.63	1.87	3.14	0.9925	0.9828	0.9287	33.01
	FV	0.11	3.02	3.54	0.46	0.9831	0.9613	0.9026	21.26

* Significant at *p* < 0.05. ^a^
*X_1_*, ultrasound irradiation power (W); *X_2_*, temperature (°C); *X_3_*, ultrasound irradiation time (min); Df, degree of freedom; SD, standard deviation; AP, adequacy precision; AC, anthocyanins; FV, flavonols.

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
