# Peer review of "Efficient Homogenization-Ultrasound-Assisted Extraction of Anthocyanins and Flavonols from Bog Bilberry (Vaccinium uliginosum L.) Marc with Carnosic Acid as an Antioxidant Additive"

_molecules, 2019, doi:10.3390/molecules24142537_

Round 1

Reviewer 1 Report

The paper „Efficient homogenization-ultrasound-assisted extraction of anthocyanins and flavonols from bog bilberry (Vaccinium uliginosum L.) marc with carnosic acid as an antioxidant additive“ deals with very interesting subject – determining the conditions of extraction method, and also using natural antioxidants to improve the process. The paper is written clearly, the results are well described. I suggest publishing the paper after minor modifications.

-In materials and method section, it would be good to describe which method you used for determining the amount of anthocyanins in certain experiment. For example, in materials and methods you described HPLC method for individual anthocyanin and individual flavonol determinations, and spectrophotometric method (pH differential method) for total anthocyanins. But you did not explain were you using HPLC method in all of those experimental conditions for extractions. In all figures (Figure 1 to 3) you show total amounts of anthocyanins while flavonols are shown as individual compounds (and probably also as total, as summ of all individual flavonols). I assume, you showed here the results of anthocyanins as total anthocyanins which you obtained by using spectrophotometric pH differential method, and the results for flavonols are the results obtained by using HPLC method. Explain better.

Also, you showed a chromatogram of anthocyanins but with no identification (Figure 4). For HPLC analysis of anthocyanins, you used only cyanidin-3-glucoside standard (3.1. materials and chemicals). So, maybe you used total peak area of anthocyanins and calculated total anthocyanins according to cyanidin-3-glucoside calibration curve? It needs to be explained. IF you just used HPLC method just to see which anthocyanins are present in the extract, that is ok, just needs to be well described.

-In Figures 1 and 2, there are no error bars on flavonols. It should be added, if possible. Since you have error bars on anthocyanin amount, I assume it is possible to do the same with flavonols.

-In describing the results from Figures 1 and 2, maybe it would be good to describe the statistical significance of differences between different experimental conditions. For example, the statistical significance of differences between extracted anthocyanins using different ethanol volume fraction (conduct some statistical tests, post hoc Tukey test???).

Reviewer 2 Report

The manuscript  proposes an optimized homogenization-ultrasound-assisted method for the extraction of anthocyanins and flavonols from bog bilberry. Considering the great importance of polyphenols especially from health-promoting point of view, the topic is really interesting. However, the paper is affected by many weaknesses in this form. First, there is a general lack of profound scientific interpretation; “results and discussion” section sounds like a merely list of observations, with no accurate comparative discussion of the findings. Specifically, in my mind it is not clear at all how you have chosen the three factors (ultrasound irradiation power, extraction temperature, and US time) from the single factor experiments results. Did you apply a Plackett Burman design for screening? Then, I have highlighted (see the file Molecules-527871_R1) a couple of concerns with regard to HPLC analyses which must be explained.

Overall, the manuscript sounds like very poorly written; therefore, in order to be able to assess its findings properly, I suggest a really deep revision of English grammar and language.

Reviewer 3 Report

Dear Authors,

having read your interesting and comprehensive manuscript I have several questions, as many issues need to be clarified prior to the publication of your research:

- in the introduction section: has any other researcher studied the optimisation of extraction of this particular species? underline the novelty of your studies.

-in the introduction section please explain the selection of these two particular antioxidants from rosemary

- line 62: what do the authors mean by 'relatively healthy'? please, indicate the toxicity of this compound and the doses applied

-line 95 the term 'volume fractions' is a wrong word in my opinion. Maybe the authors should rewite this section using the 'ethanol percentage' term to make the paragraph more clear for the readers.

- Figure 1 and 2: I miss the crude data - the numbers and their SD on the graphs to better judge the actual differences between the samples. 

- how much of CA may be added to the extracted sample not to induce toxicity? could the authors comment on that?

- line 154: how to explain that there is no linearity in these results? 80>45>100....

- the weakest part of the manuscript includes the malterials and methods section. please, correct it in a way that other researchers can repeat your studies. divide the mathods section into clear subsections: like extraction parameters, homogenisation protocol, and so on. please, include all tested conditions: for example writing about dissolving in aqueous alcohol of a certain volume (line 294) mention an exact range of used volumes and the number of checked solutions. Show the range of time, duration of extraction and in this way characeterise all parameters adjusted by the authors in this study. now we can see the results of your study but cannot read about how you planned your experiment

- line 273: who authenticated the samples? do the authors store a voucher specimen in their unit?

- line 313: have the authors optimised this method on their own? in the analysis of anthocyans the gradient of acetonitrile reaches only 25%. how about other compounds with a smaller polarity, like flavonoids? with this method they stay in the column...in your studies have you obtained another chromatogram for this method towards anthocyanins with a wider range of acetonitrile addition to show in this paper? after several injections the pressure on the column certainly increased...

-line 328: to line 331: move this section to the results, as these are results

- line 338: it is not clear - why the authors do not relate the quantity to the mass of the dried extract? please, comment on that

- line 350: has this methodology been published before? please, give a citation here

Reviewer 4 Report

The manuscript entitled " Efficient homogenization-ultrasound-assisted extraction of

anthocyanins and flavonols from bog bilberry (Vaccinium uliginosum L.) marc

with carnosic acid as an antioxidant additive" investigated the optimum conditions for the extraction of anthocyanins and flavonols from bog bilberry (Vaccinium uliginosum L.) marc. Although the experiments appear to be well planned, results seem interesting and correct, the ideas and methods are standard and the results are solid, however, the topic does not contain new ideas and the discussion section is standard and not surprising. In my mind, this article is not acceptable for publication in Molecules.

Author Response

Dear Reviewer,

Thank you for your comments concerning our manuscript entitled “Efficient homogenization-ultrasound-assisted extraction of anthocyanins and flavonols from bog bilberry (Vaccinium uliginosum L.) marc with carnosic acid as an antioxidant additive” (ID: molecules-527871). Those comments are all valuable and very helpful for revising and improving our paper, as well as the important guiding significance to our researches. We have studied comments carefully and have made correction which we hope meet with approval. All revisions made to the manuscript have been marked up with the "Track Changes" function in the revised manuscript. The main corrections in the paper and the responds to the reviewer’s comments are as following:

The manuscript entitled “Efficient homogenization-ultrasound-assisted extraction of anthocyanins and flavonols from bog bilberry (Vaccinium uliginosum L.) marc with carnosic acid as an antioxidant additive” investigated the optimum conditions for the extraction of anthocyanins and flavonols from bog bilberry (Vaccinium uliginosum L.) marc. Although the experiments appear to be well planned, results seem interesting and correct, the ideas and methods are standard and the results are solid, however, the topic does not contain new ideas and the discussion section is standard and not surprising. In my mind, this article is not acceptable for publication in Molecules.

Response:

Thank you very much for the comments.

Maybe we haven't clarified the innovation of this paper. The innovation of this paper is that most of the previous studies have extracted anthocyanins and flavonols from bog bilberry fruits, while we extracted anthocyanins and flavonols from fruit marc which was originally a waste. In addition, the application of carnosic acid as an antioxidant in the extraction of anthocyanins and flavonoids has not been reported before. This paper reported that carnosic acid has played a good role in the extraction of anthocyanins and flavonoids from bog bilberry marc.

The findings of this study were further interpreted scientifically and discussed accurately in the section of “Results and Discussion”. The details can be found in the revised manuscript.

We appreciate for Editors and Reviewers’ warm work earnestly, and hope that the correction will meet with approval.

Once again, thank you very much for your comments.

Sincerely yours,

Lei Yang

Round 2

Reviewer 2 Report

I have to say the authors have made a really good work. Actually, the manuscript seems very improved in this new version. Just some minor corrections need to make it fully acceptable (see the attached file molecules-527871-R2).

Author Response

Dear Reviewer,

Thank you for your comments concerning our manuscript entitled “Efficient homogenization-ultrasound-assisted extraction of anthocyanins and flavonols from bog bilberry (Vaccinium uliginosum L.) marc with carnosic acid as an antioxidant additive” (ID: molecules-527871). Those comments are all valuable and very helpful for revising and improving our paper, as well as the important guiding significance to our researches. We have studied comments carefully and have made correction which we hope meet with approval. All revisions made to the manuscript have been marked up with the "Track Changes" function in the revised manuscript. The main corrections in the paper and the responds to the reviewer’s comments are as following:

I have to say the authors have made a really good work. Actually, the manuscript seems very improved in this new version. Just some minor corrections need to make it fully acceptable (see the attached file molecules-527871-R2).

Question 1:

In my first revision I meant you should substitute the words already present in the title with new ones. Such as: antioxidant capacity, polyphenol, fruit marc, natural antioxidant.

Response:

Thank you for the comments. We have substituted the words already present in the title with “antioxidant capacity; polyphenol; fruit marc; natural antioxidant” in line 30.

Question 2:

Move lines 134-137 before line 128.

Response:

Thank you very much for the good suggestion. We have moved lines 95-97 before line 91.

Question 3:

Line 322-323, sorry but the meaning of what you want to say is not clear yet at all.

Response:

We are very sorry for the unclear expression, and we have rephrased the sentence as follows: “That is, 16 mL/g of liquid-solid ratio had almost completely dissolved the effective components” in line 156-157.

Question 4:

Line 374-376, sorry but the two sentences appear antithetic in this form. Please, rephrase more clearly.

Response:

We are very sorry for the unclear interpretation. We have rephrased the sentence as follows: “the higher the frequency was, the smaller the cavitation bubbles and the weaker the cavitation intensity.” in line 192-193.

Once again, thank you very much for your comments and suggestions.

Sincerely yours,

Lei Yang

Reviewer 3 Report

Dear Authors,

thank you very much indeed for your corrections introduced to the text.

I still have two minor remarks:

- in the materials and methods> HPLC of anthocyanins section - please add a sentence, where you state, that the authors flushed the apparatus and the HPLC column with acetonitrile or methanol (as you state in the detailed responses to my previous comments). It will look better methodologically

- concerning the graphs, i think, that we misunderstood each other a bit. I would be very grateful, if you could add specific numeral values to your figures, e.g. to write above each bar on the chart a numeral value, e.g. 1.4% or 1.6% - just to know exactely the numeric values obtained in your studies.

Author Response

Dear Reviewer,

Thank you for your comments concerning our manuscript entitled “Efficient homogenization-ultrasound-assisted extraction of anthocyanins and flavonols from bog bilberry (Vaccinium uliginosum L.) marc with carnosic acid as an antioxidant additive” (ID: molecules-527871). Those comments are all valuable and very helpful for revising and improving our paper, as well as the important guiding significance to our researches. We have studied comments carefully and have made correction which we hope meet with approval. All revisions made to the manuscript have been marked up with the "Track Changes" function in the revised manuscript. The main corrections in the paper and the responds to the reviewer’s comments are as following:

Dear Authors,

thank you very much indeed for your corrections introduced to the text.

I still have two minor remarks:

Question 1:

In the materials and methods> HPLC of anthocyanins section - please add a sentence, where you state, that the authors flushed the apparatus and the HPLC column with acetonitrile or methanol (as you state in the detailed responses to my previous comments). It will look better methodologically.

Response:

Thank you very much for the good suggestion. We have added a sentence as follows: “(The apparatus and the HPLC column were flushed with acetonitrile or methanol, and then the column was balanced with mobile phase until the baseline of the chromatogram getting stable before each injection.)” in line 355-357.

Question 2:

Concerning the graphs, i think, that we misunderstood each other a bit. I would be very grateful, if you could add specific numeral values to your figures, e.g. to write above each bar on the chart a numeral value, e.g. 1.4% or 1.6% - just to know exactely the numeric values obtained in your studies.

Response:

Thank you very much for the valuable comment. We have added specific numeral values to Figures 2-4.

Once again, thank you very much for your comments and suggestions.

Sincerely yours,

Lei Yang

Reviewer 4 Report

The manuscript entitled “Efficient homogenization-ultrasound-assisted extraction of anthocyanins and flavonols from bog bilberry (Vaccinium uliginosum L.) marc with carnosic acid as an antioxidant additive” investigated the optimum conditions for the extraction of anthocyanins and flavonols from bog bilberry (Vaccinium uliginosum L.) marc on a single-factor experimental basis and response surface methodology. It can be accepted in Molecules with some points.

Line 86-87 Ultrasound-assisted extraction (UAE) has the advantages of high efficiency and reproducibility, short extraction time, low solvent consumption, simple manipulation and high level of automation [24-27].

27 Feng, S.; Luo, Z.; Tao, B.; Chen, C. Ultrasonic-assisted extraction and purification of phenolic compounds from sugarcane (Saccharum officinarum L.) rind. LWT-Food Science and Technology201560: 970-976

88-89  The ultrasound mechanism is conducive to greatly improving the solvent infiltration into cellular materials, and the cavitation effects promote the release of cell inclusion into the bulk media [28-29].

29 Huang, H.; Xu, Q.; Belwal, T.; Li, L.; Aalim, H.; Wu, Q.;  Duan, Z.; Zhang, X.; Luo, Z. Ultrasonic impact on viscosity and extraction efficiency of polyethylene glycol: A greener approach for anthocyanins recovery from purple sweet potato. Food Chemistry, 2019, 283: 59-67

Author Response

Dear Reviewer,

Thank you for your comments concerning our manuscript entitled “Efficient homogenization-ultrasound-assisted extraction of anthocyanins and flavonols from bog bilberry (Vaccinium uliginosum L.) marc with carnosic acid as an antioxidant additive” (ID: molecules-527871). Those comments are all valuable and very helpful for revising and improving our paper, as well as the important guiding significance to our researches. We have studied comments carefully and have made correction which we hope meet with approval. All revisions made to the manuscript have been marked up with the "Track Changes" function in the revised manuscript. The main corrections in the paper and the responds to the reviewer’s comments are as following:

The manuscript entitled “Efficient homogenization-ultrasound-assisted extraction of anthocyanins and flavonols from bog bilberry (Vaccinium uliginosum L.) marc with carnosic acid as an antioxidant additive” investigated the optimum conditions for the extraction of anthocyanins and flavonols from bog bilberry (Vaccinium uliginosum L.) marc on a single-factor experimental basis and response surface methodology. It can be accepted in Molecules with some points.

Question 1:

Line 86-87: Ultrasound-assisted extraction (UAE) has the advantages of high efficiency and reproducibility, short extraction time, low solvent consumption, simple manipulation and high level of automation [24-27].

Response:

Thank you vary much for the suggestion. We have added the appropriate literature in line 81. ([27] Feng, S.; Luo, Z.; Tao, B.; Chen, C. Ultrasonic-assisted extraction and purification of phenolic compounds from sugarcane (Saccharum officinarum L.) rind. LWT - Food Sci. Technol. 2015, 60, 970976.).

Question 2:

Line 88-89: The ultrasound mechanism is conducive to greatly improving the solvent infiltration into cellular materials, and the cavitation effects promote the release of cell inclusion into the bulk media [28-29].

Response:

Thank you very much for the comment. We have added the appropriate literature in line 83. ([29] Huang, H.; Xu, Q.; Belwal, T.; Li, L.; Aalim, H.; Wu, Q.; Duan, Z.; Zhang, X.; Luo, Z. Ultrasonic impact on viscosity and extraction efficiency of polyethylene glycol: A greener approach for anthocyanins recovery from purple sweet potato. Food Chem. 2019, 283, 5967.).

Once again, thank you very much for your comments and suggestions.

Sincerely yours,

Lei Yang